



# Unsupervised image velocimetry for automated computation of river flow velocities

Matthew T. Perks[1], Borbála Hortobágyi[1,2], Nick Everard[3], Susan Manson[4], Juliet Rowland[5], Andrew Large[1], and Andrew J. Russell[1]

[1]School of Geography, Politics and Sociology, Newcastle University, Newcastle upon Tyne, United Kingdom
[2]UMR5600 EVS, CNRS, Lyon, France
[3]UK Centre for Ecology and Hydrology: Wallingford, United Kingdom
[4]Flood and Coastal Risk Management, Environment Agency, Crosskill House, Mill Land, Beverley
[5]Environment Agency, Manley House, Kestrel Way, Exeter

**Correspondence:** Matthew T. Perks (matthew.perks@newcastle.ac.uk)

**Abstract.** Accurate, long-term, measurements of river flow are imperative for understanding and predicting a broad range of fluvial processes. Modern technological advances are enabling the development of new solutions that are tailored to manage water resources and hazards in a variety of flow regimes. This study appraises the potential of freely available image velocimetry software (KLT-IV) to provide automatic determination of river surface velocity in an unsupervised workflow. In this research, over 11,000 videos are analysed, and these are compared with 1-D velocities derived from 303 flow gauging measurements obtained using standard operating procedures. This analysis was undertaken at a complex monitoring site with a partial view of the channel with river flows spanning nearly two orders of magnitude. Following image velocimetry analysis, two differing approaches are adopted to produce outputs that are representative of the depth-averaged and cross-section averaged flow velocities. These approaches include the utilisation of theoretical flow field distributions to extrapolate beyond the field of view, and an index-velocity approach to relate the image-based velocities to a section averaged (1-D) velocity. Analysis of the section-averaged velocities obtained using KLT-IV, compared to traditional flow gauging, yields highly significant linear relationships ($r^2$ = 0.95-0.97). Similarly, the index-velocity approach enables KLT-IV surface velocities to be precisely related to the section-averaged velocity measurements ($r^2$ = 0.98). These data are subsequently used to estimate river flow discharge. When compared to reference flow gauging data, $r^2$ values of 0.98 to 0.99 are obtained (for a linear model with intercept of 0 and slope of 1). KLT-IV offers an attractive approach for conducting unsupervised flow velocity measurements in an operational environment where autonomy is of paramount importance.

## 1 Introduction

Accurate hydrological data are fundamental to enable advances in understanding the physical processes occurring in river systems (e.g. sediment entrainment, transport, and channel change), to drive hydraulic models that predict the extent of floods, and to provide flood warnings to the public (McMillan et al., 2017; Tauro et al., 2018a). However, classical approaches for the determination of key hydrological variables such as river discharge are costly to maintain and require investment of significant





resources (Fekete and Vörösmarty, 2007). In the absence of measuring structures (e.g. weirs, flumes), continuous time-series of flow discharge are most commonly generated through acquisition of episodic, paired observations of river stage and flow discharge, from which a stage-discharge relation can be computed with continuous discharge data being subsequently generated

as a function of river stage (Kiang et al., 2018). As part of this workflow, velocity measurements are often carried out using an acoustic Doppler current profiler (aDcp), or current meter (Herschy, 2014). However, there is demand for lower cost solutions, and development of techniques that are more readily applicable in periods of high flow when the use of standard approaches may not be possible, or induce elevated uncertainty (Kidson and Richards, 2005; Di Baldassarre and Montanari, 2009).

One approach that has gained interest in recent years is the computation of water surface velocity through image-processing

techniques, and its subsequent conversion to a mean section velocity (Jolley et al., 2021). Measurements of surface velocity can be achieved through the application of existing algorithms that may be broadly categorised as: particle tracking velocimetry (PTV) (Brevis et al., 2011; Tauro et al., 2017); large scale particle image velocimetry (LSPIV) (Fujita et al., 1998; Muste et al., 2008); or space-time image velocimetry (STIV) (Fujita et al., 2007). These techniques were initially employed to acquire flow measurements from fixed stations (Bradley et al., 2002; Hauet et al., 2008; Stumpf et al., 2016), or temporary ground stations

(Jodeau et al., 2008; Kim et al., 2008; Dramais et al., 2011). These have subsequently been applied to imagery acquired from uncrewed aerial systems (e.g., Lewis et al., 2018; Masafu et al., 2022), and mobile phones (e.g., DischargeApp; Peña-Haro et al., 2021).

More recently, optical flow algorithms have been successfully applied as a means of computing surface velocities from fixed cameras (Tauro et al., 2018b; Lin et al., 2019; Khalid et al., 2019) and uncrewed aerial systems (Perks et al., 2016). These

computer vision algorithms automatically identify pixels that are distinct from their neighbours, and these distinct features can be iteratively tracked through a sequence of images through varying applications of the Lucas-Tomasi algorithm. These approaches are computationally very efficient and capable of performing analysis up to two orders of magnitude faster than traditional PIV and PTV approaches (Tauro et al., 2018b). Benchmarking studies have found these algorithms to produce velocity measurements comparable to current meter data (Tauro et al., 2018b), aDcp data (Pearce et al., 2020), and produce

more reliable measurements than traditional image velocimetry approaches in the laboratory and field when using thermal cameras and thermal tracers (Lin et al., 2019).

However, regardless of the tracking algorithm adopted, the application of image velocimetry in a continuous, automated, and unsupervised workflow for the purpose of sensing river flows continues to pose a challenge. Generally, in order to ensure high quality measurements under all conditions, image-based approaches benefit from a homogeneous distribution of tracers

on the surface, which is seldom the case when sensing complex natural fluvial systems. But more specifically, each approach requires parameterisation which can be difficult to define for all flow and environmental conditions to which the system may be exposed. For example, in the case of LSPIV, frame extraction rates, interrogation and search areas should be appropriately defined for optimal performance, and whilst the latter has been improved through application of spatio-temporally-adaptive search areas (Fleit and Baranya, 2019), multiple passes and deforming windows (e.g., Thielicke and Sonntag, 2021), they are

still a critical consideration. Similarly, despite recent advances in the development of STIV, the automatic detection of the main orientation of texture in instances of low-quality STIs remains problematic (Wang et al., 2024).





There are, however, notable exceptions in the development and application of automated and unsupervised workflows for sensing river flow velocities using image sequences. Hauet et al. (2008) deployed an experimental LSPIV-based system for 23 months on the Iowa River and produced exemplary results by analysing image pairs acquired at 1-s intervals with constant

interrogation and search areas. Ran et al. (2016) subsequently deployed an automated Raspberry Pi-based LSPIV system for continuous monitoring of flood flow measurements in a mountainous catchment where spot velocity measurements indicated errors of generally less than 8%. More recently, another cross-correlation-based approach was presented by Photrack AG. Their system (DischargeKeeper) was shown to be capable of acquiring continuous image velocimetry results with a high-degree of accuracy in three specific case-studies (Peña-Haro et al., 2021). The application of optical flow algorithms in the context of

continuous and automated velocimetry workflows has also garnered interest due to their relative insensitivity to parametrisation (e.g., Pearce et al., 2020; Tosi et al., 2020). This approach was used by Hutley et al. (2023), where they presented their computer vision stream gauging (CVSG) system which uses the Farneback algorithm to solve the optical flow equation for determining surface flow fields. In application of the CVSG system on the Tyenna River ($Q\approx$1-20 $m^3$ $s^{-1}$) with measurements made at a distance of 5.9 to 7.3m from the camera, results were strong under all conditions (with Nash–Sutcliffe efficiency (NSE) values

of between 0.91-0.97). However, in their deployment on the larger Paterson River ($Q\approx$0-600 $m^3$ $s^{-1}$) with measurements made at a distance of 0 to 22.5m from the camera, surface flow fields were, in some instances, poorly resolved. Hutley et al. (2023) attributed this to challenging water surface textures, the distance from the camera to the water surface, and the channel cross-section approaching the eye level of the camera at higher observed flows. Several of these issues are likely to be intermittently present during continuous deployments of camera systems for sensing flow fields, and appropriate methods for mitigating these

are required. In the case of both Peña-Haro et al. (2021) and Hutley et al. (2023), challenges that resulted in the flow field being poorly resolved are countered through the application of algorithms that over time 'learn' the shape of surface velocity profiles for a specific site, with this being subsequently applied to any instances of missing data in the sensed cross-section.

Similarly to Hutley et al. (2023), in the research paper we present here, we encounter challenges in the automated sensing of water surface velocities across the full range of flow conditions observed (flow discharge of 1.7 to 145$m^3$ $s^{-1}$). Through

application of a computer vision based workflow implemented within KLT-IV (Perks, 2020), we assess the potential for the automatic determination of river flow velocities in a complex setting where there is a partial view of the river channel. In order to achieve this, we have the following research objectives: (i) to examine how 1-D velocity measurements derived from traditional flow gauging techniques compare with measurements obtained using KLT-IV; (ii) to examine and quantify the effects of data driven fitting approaches on subsequent section average velocity estimates; and (iii) to assess whether an index-velocity

approach can be applied to convert distributed surface velocities obtained using KLT-IV to a section-averaged velocity.

## 2  Methods

### 2.1  Experimental Site

At the site of the field experiment (50.479247° N, -3.761540° E; Figure 1), the River Dart is a 247.6km$^2$, rapidly responding catchment, characterised by steep relief and a long-term annual rainfall of 1771mm. It is a predominantly alluvial channel,





with some exposed bedrock, and a width of approximately 25m under normal flow conditions. The cross-section is stable with

repeat surveys in 2010 and 2019 showing a 5% variation in cross-sectional area across the full range of flows experienced.

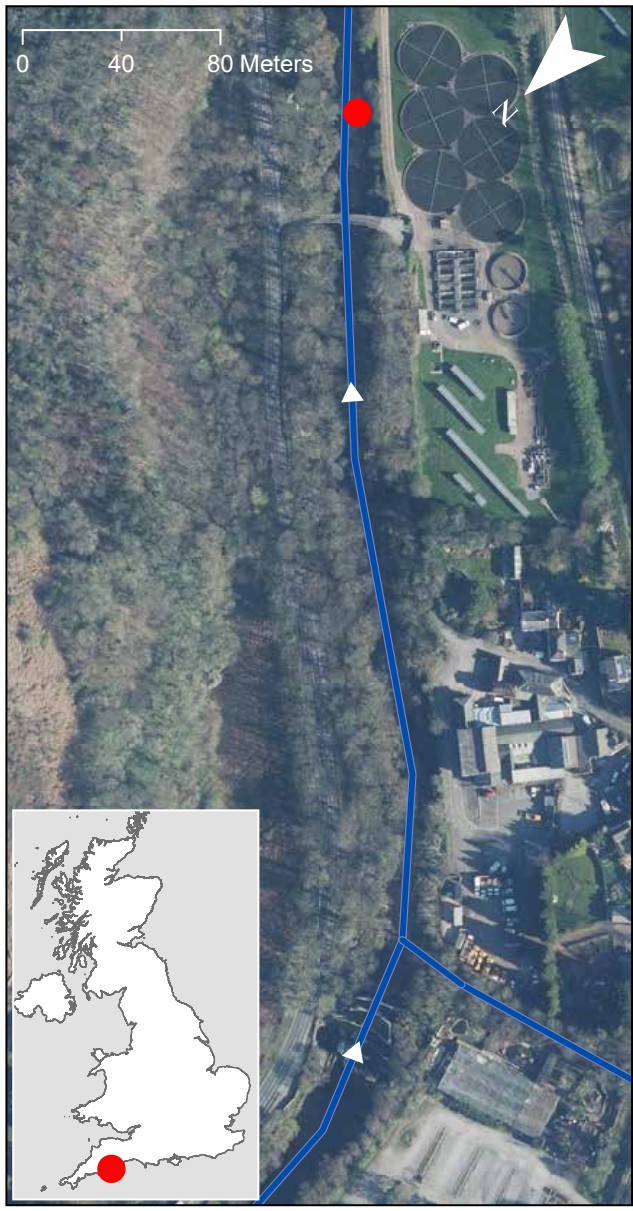

**Figure 1.** Location of the field experiment on the River Dart at Austins Bridge (red circle). River network and flow direction are shown by
the blue lines and arrows respectively. Aerial imagery provided by EDINA Aerial Digimap Service (2022). Inset map shows the monitoring
location in the context of the UK.





## 2.2 Image Acquisition

The image acquisition hardware consists of an obliquely mounted Hikvision DS-2CD2T42WD-I8 6mm IP camera connected via ethernet cable to a Raspberry Pi Model 3B. The camera is located on the true right bank at a height of 4.75m and 2.55m above the water surface at the minimum and maximum observed river stage respectively. The camera is mounted at an oblique angle of 77 degrees from nadir. Despite the adoption of an oblique camera angle, the camera does not observe between 4.6m and 8.6m of the water surface (in the near distance). This accounts for between 21–26% of the cross-section across the full range of flow conditions observed in this study (Figure 2). The images are acquired at a resolution of 1920 x 1080px at a rate of $19.99 \pm 0.5$Hz (95% confidence interval; Perks, 2024c). These are of 10-s duration and collected at 15-minute intervals. In this analysis we utilise videos obtained during daylight hours between March 2018 and March 2019. Low-light, and night-time imagery (determined when infra-red sensing was triggered) were removed due to inconsistent visibility of surface features. The first 3-s of each recording were eliminated from analysis as these frames experienced compression and frame rate issues.

## 2.3 Image Calibration

Prior to image analysis, a distorted camera model is established for the site (Messerli and Grinsted, 2015; Perks et al., 2016). This is required to enable transformation from image coordinates to geographic coordinates. The requirements of this model are: (i) intrinsic parameters of the camera (either measured or estimated); (ii) surveyed location of ground control points (GCPs); (iii) location [x, y, z] and orientation of the camera; and (iv) the known height of the water surface being sensed. In order to calculate the intrinsic parameters of the camera model (i.e. radial and tangential distortion coefficients, camera focal length, and image centre parameters), geometric calibration was conducted using a 841 x 1189mm checker-board pattern and the Camera Calibrator App within Matlab 2019b. A total of 40 images were used in this process which resulted in a mean re-projection error of 0.61px. The pixel coordinates of nine GCPs across the camera's field-of-view were obtained from imagery acquired by the Hikvision camera and the geographical coordinates of the GCPs were determined through acquisition of a high spatial resolution point cloud using a Leica MS50 multi-station. The dynamic nature of the water surface elevation over time is taken into account by automatically setting the water surface representation ($z_{m_{[x,y]}}$). This was achieved through the definition of the water surface elevation from the point cloud at the time of the survey ($S_{initial}$), the river stage at the time of survey ($h_{initial}$), and continuous river stage measurements performed using a float and counterweight shaft encoder at 15-minute intervals ($h$):

$$z_{m_{[x,y]}} = S_{initial} + (h - h_{initial}) \tag{1}$$

The camera location and view direction (yaw, pitch, and roll) was initially estimated using the point-cloud survey. These characteristics were then defined as free parameters and optimized to minimize the square projection error of the GCPs using a modified Levenberg–Marquardt algorithm (Fletcher, 1971).

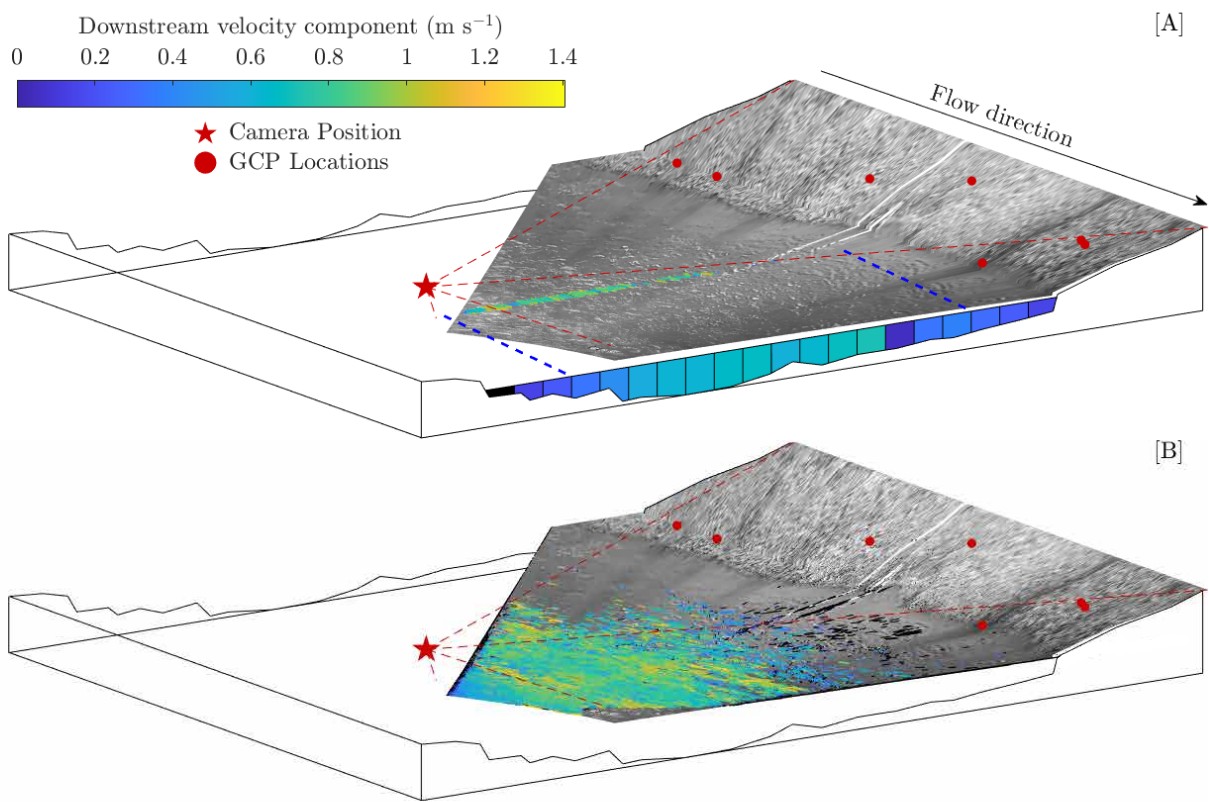

**Figure 2.** Schematic illustrating the monitoring station setup and the camera's partial view of the river cross-section. Cross-section data is presented up to the maximum observed river level. Red dashed lines illustrate the camera's field of view. The image presented is an orthophoto produced from footage acquired on 29[th] December 2018 at 13:00GMT when the flow discharge was $\approx 12\text{m}^3\text{ s}^{-1}$, and section averaged flow velocity was $\approx 0.54\text{m s}^{-1}$. Vectors represent the direction and velocity of [A] only those features that pass through the cross-section of interest; and [B] all tracked features within the region of interest. Vectors coloured black are trajectories that have been filtered. In the application of theoretical flow field distributions (Figure 2A; Section 2.5.1), surface velocity data is converted to a depth-averaged before being binned into one of 20 equal-width cells enabling the cell-averaged velocity to be obtained. Blue dashes represent the spatial extent of the detected surface features and extrapolation of the flow field is required beyond this extent. The foundation of the velocity index approach (Figure 2B; Section 2.5.2), is that the average surface velocity from across the field of view can be linearly related to the 1-D velocity.



## 2.4 Image Processing

Prior to image velocimetry analysis, image sequences were orthorectified using the optimised camera model described in Section 2.3 and exported with a pixel size of 0.01m x 0.01m. Subsequently, these were subject to pre-processing to enhance the visibility of surface features. Specifically, high-frequency components of the orthorectified imagery were enhanced through
application of a high-pass filter with kernel size of 32px x 32px. This was achieved by calculating a low-passed version of the original image and subtracting it from the original (Thielicke and Sonntag, 2021). Additional inputs were also defined including the region-of-interest to ensure that areas of the image containing on screen display information (e.g. timestamp) and areas consisting of artificial noise (e.g. tree branches) were excluded from the analysis. The primary flow direction was also defined to enable both the primary and secondary components to be computed.

The workflow for image velocimetry analysis consists of the automatic detection of naturally occurring surface water features using a minimum eigenvalue algorithm (Shi and Tomasi, 1994). Features were subsequently tracked from frame-to-frame using a MATLAB implementation of the Kanade–Lucas–Tomasi algorithm (Lucas and Kanade, 1981; Tomasi and Kanade, 1991; Shi and Tomasi, 1994; Perks, 2020). The adopted approach tracks windows of features of 31px x 31px in size, from which an affine motion field is generated to assign velocities to different points within the window. Instances where pixel motion of
lengths greater than the pre-defined window size are handled through the use of pyramid levels (Bouguet, 2000). This approach effectively down-samples the original image by a factor of two between each pyramid level; three pyramid levels were used in our analysis. The lowest pyramid level provides an initial estimate of the pixel displacement using the coarsest imagery. This is then refined in a recursive fashion through the pyramid levels up to the original image resolution (Bouguet, 2000).

Evaluation of feature tracking success is achieved through implementation of a forward-backward error propagation scheme.
Firstly, forward trajectories are computed and stored based on apparent feature movement from the first to the last frame in the sequence. These trajectories are then compared with those derived by backward tracking the feature from the last to the first frame in the sequence. If the difference between the trajectories exceeds one pixel, the trajectory is considered incorrect and removed from analysis (Kalal et al., 2010).

Features are tracked over a period of 0.50s (ten frames) from which the start and finish positions (in metric units) are stored.
These are converted to displacement rates $(\text{m s}^{-1})$, and broken down into their downstream and secondary velocity components. Two post-processing approaches are implemented to filter spurious vectors, namely the removal of: (i) vectors that deviate from the user-defined flow-line by $\geq 45°$; and (ii) vectors with a (user-defined) displacement of $< 0.1\text{m s}^{-1}$. The former acts as to filter those vectors that are likely spurious based on their direction, whereas the latter filters objects that are close to being stationary. These typically represent objects located on the channel margins, or reflections on the water surface (e.g. bank-side
reflections observed in Figure 2B).

## 2.5 Experimentation

Upon the reconstruction of the surface flow velocity field, it is common for these observations to be converted to data that describe the depth-averaged velocity at multiple points in the cross-section. This forms the basis for the widely adopted velocity-





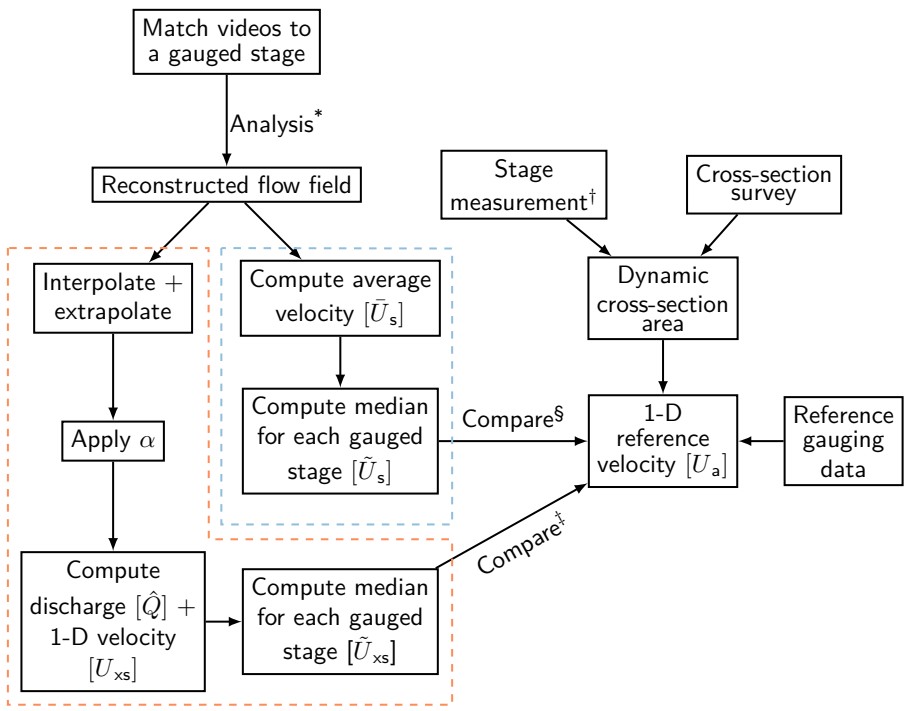

**Figure 3.** Schematic diagram illustrating the workflow of the data analysis as described in Section 2.5. * Analysis settings are described in Section 2.1-2.4. Items within the red box relate to the methods presented in Section 2.5.1, and items in the blue box are related to Section 2.5.2. † Derivation of river stage measurements in the local coordinate system are shown in Equation 1. ‡ Results are presented in Section 3.1 and § results are presented in Section 3.2.

area method of flow discharge calculation (Herschy, 2014). However, when applying image velocimetry techniques in natural

fluvial settings, it may not be possible for equidistant velocity measurements to be extracted from across the full channel width (e.g. Peña-Haro et al., 2021; Hutley et al., 2023). Gaps in measurements can be caused by variability in lighting, inhomogeneous tracer distribution, reduced pixel resolution of the far-field, or the field-of-view failing to capture the active channel width (as evident in Figure 2A). Several approaches have been adopted to account for these failings, which commonly involve either interpolation between cells of missing data, or extrapolation beyond the observed through utilisation of theo-

retical flow field distributions (Leitão et al., 2018; Le Coz et al., 2010; Fulford and Sauer, 1986). Conversely, an alternative approach for converting the information contained within the surface velocity flow field to a flow velocity that is representative of the cross-section is through the development of empirical relationships between observed and reference observations (e.g. adoption of an index-velocity approach (Levesque and Oberg, 2012), or application of entropy theory (Chiu, 1989; Moramarco and Singh, 2010; Vyas et al., 2024)). Here we test these two distinct approaches, which are described in the following sections

and conceptualised in Figure 3.





### 2.5.1 Utilisation of theoretical flow field distributions

In order to interpolate between and extrapolate beyond the extent of the surface velocity field, one of several assumptions about the flow field may be employed. Here we adopt three approaches: an assumption that the Froude number is constant within a cross-section (Le Coz et al., 2010; Fulford and Sauer, 1986), and the adoption of quadratic, and cubic polynomials (Leitão et al., 2018). Upon application of these techniques, average velocities for 20 segments of equal sizes are established (Figure 2A). This is then converted to a depth-averaged velocity using a conversion factor ($\alpha$) of 0.87. This site-specific value was obtained following analysis of 60 aDcp transects acquired between 2009 and 2018 (Perks, 2024b). Subsequently, the average cell velocity is multiplied by the cell area to give the unit-discharge of each cell from which the flow discharge estimate at the cross-section of interest is obtained ($Q$). This discharge value is divided by the wetted cross-sectional area $A_s$ to provide the image-based section-averaged (or 1-D) velocity ($U_{\mathrm{xs}}$).

Our analysis focuses on the comparison between the section-averaged (1-D) velocities obtained by KLT-IV ($U_{\mathrm{xs}}$) and the section-averaged (1-D) velocities derived from reference observations $U_a$. The reference flow measurements span the range of 1.7–145m$^3$ s$^{-1}$. The lower end of of this range represents flow conditions that approximate the long-term 95% exceedance value, whereas the highest flows analysed are of greater magnitude than the long-term 1% exceedance value (UK CEH, 2024). To calculate the (1-D) velocities derived from reference observations we simply divide the measured discharge by expected cross-sectional area based on a combination of the measured river stage and geodetic surveys. This approach is adopted even in the case where aDcp gauging measurements act as the reference flow data due to the potential for bias in cross-section measurements from aDcp data (discussed in Section 4.2). For each of the reference measurements, videos acquired at the same river stage ($\pm 0.01$m) are selected and the median of the 1-D velocities $\tilde{U}_{\mathrm{xs}}$ are used for further analysis. Additionally, a comparison between the KLT-IV depth-averaged velocities and Sontek RiverSurveyor M9 aDcp velocities are presented for eight flow gauging measurements. The choice of video to compare with the aDcp gauging data was based on the selection of the image-based 1-D velocity ($U_{\mathrm{xs}}$) that corresponds most closely with the median value ($\tilde{U}_{\mathrm{xs}}$) for the same flow stage as the aDcp data was acquired ($\pm 0.01$m).

### 2.5.2 Index-velocity

Using a combination of traditional flow-gauging measurements and KLT-IV derived velocities, relationships between the mean surface velocity from across the camera's field of view (index velocity, $\bar{U}_s$) and the observed section-averaged velocity (1-D velocity, reference velocity, $U_a$) can be generated. This was initially calculated for the calibration period (March - June 2018), and then applied and tested for the validation period (July 2018 - March 2019). During these time periods, videos were selected for analysis that were acquired at river levels coinciding with those of the available flow gauging measurements ($\pm 0.01$m). For each video the average surface velocity $\bar{U}_s$ was calculated and the median of the $\bar{U}_s$ values for each river level corresponding to a reference flow gauging measurement is calculated [$\tilde{U}_s$]. The derived $\tilde{U}_s$ were compared with the 1-D velocity derived from flow gauging measurements complied between 1981 and 2018 by the UK Environment Agency through application of least squares regression between the two variables (Section 3.2). To calculate the (1-D) velocities derived from reference





observations we again divide the measured discharge by expected cross-sectional area based on a combination of the measured
river stage and geodetic surveys.

Given the required calibration step, the sensitivity of the calibration to the number of $U_\mathrm{a}$ observations is evaluated. The
number of measurements used in the calibration ($n$) ranged from 1–50. For each simulation a random 1-D reference velocity
$U_\mathrm{a}$ was selected. Simultaneously, one $\bar{U}_\mathrm{s}$ value, obtained at the same river stage as the reference gauging, was also sampled
with replacement. For each pairing, the difference between $\bar{U}_\mathrm{s}$ and $U_\mathrm{a}$ is calculated and the mean percent difference $[D_k]$ is
calculated as the number of flow gauging measurements is iteratively increased ($n$ = 1:50). These simulations were executed
100,000 times to account for the effects of sample size.

$$D_k = \frac{1}{n}\sum_{i=1}^{n}(\bar{U}_{s_i} - U_{a_i})/U_{a_i} \cdot 100 \qquad (2)$$

## 3 Results

### 3.1 Velocity reconstruction

1-D velocities determined from Environment Agency gauging measurements range from 0.12 to 2.33m s$^{-1}$ with a median
of 0.39m s$^{-1}$. River flows with lower velocities are more frequently gauged than are periods of higher velocity, resulting in
the gauged data being positively skewed (skewness value of 1.2). 24% of the gauged flows have 1-D velocities in excess of
1m s$^{-1}$, and 5% in excess of 2m s$^{-1}$. This is indicative of the challenges associated with acquiring flow gauging data using
standard operating procedures under high-flow conditions. When we consider the relationships between $\tilde{U}_\mathrm{xs}$ and $U_\mathrm{a}$, we can
identify strong linear relationships (r$^2$ = 0.95–0.96), with the linear models having intercepts ranging from -0.01–0.032 and
slopes ranging from 1.029–1.091 (Figure 4). The performance of the quadratic model (Figure 4A) and cubic model (Figure
4B) for interpolation and extrapolation of missing data within the cross-section are comparable, whereas the constant Froude
approach is in closer agreement with $\tilde{U}_\mathrm{a}$ (Figure 4C). The model coefficients presented indicate that for each of the fitting
methods adopted, $\tilde{U}_\mathrm{xs}$ generally underestimates relative to $U_\mathrm{a}$. However, this relationship is not constant with some variability
observed. Taking the best performing model (constant Froude approach) as an example, at reference velocities of between 0.12
to 0.5m s$^{-1}$, 1-D estimates differ from the reference values by +12.5%, -20% between 0.5 to 1.0m s$^{-1}$, -10% between 1.0 and
2.0m s$^{-1}$, and -0.8% for reference velocities in excess of 2.0m s$^{-1}$.

Comparison of velocity profiles generated by Sontek River Surveyor M9 aDcp to those produced using KLT-IV provides
further insight into the variation between modern hydrometric methods (ISO 24578:2021) and the image-based approach.
Overall, the mean absolute percentage error of the 1-D velocity estimates derived from the available Sontek M9 aDcp data
and IV outputs is 9.8%, with the aDcp and KLT-IV profiles being most similar under high flow conditions ($Q$= 48–101m$^3$ s$^{-1}$;
Figure 5 A–B). For most examples, the area of the cross-section most proximal to the camera (distance of 10-20m from left
bank) is generally in close correspondence with the aDcp data. However, there are exceptions to this. Under the highest flow
conditions (shown in Figure 5 A–B), the velocities in the near field of the imagery are not reconstructed accurately, with





overestimations of up to 50% (in the case of A) and underestimations of up to 75% (in the case of B). In addition, whilst we observe that the constant Froude number extrapolation procedure works well in the majority of cases (Figure 5 A–E), there are significant overestimations in the far-field of the imagery observed under low-flow conditions ($Q$= 3–7m$^3$ s$^{-1}$; Figure 5 F–H).



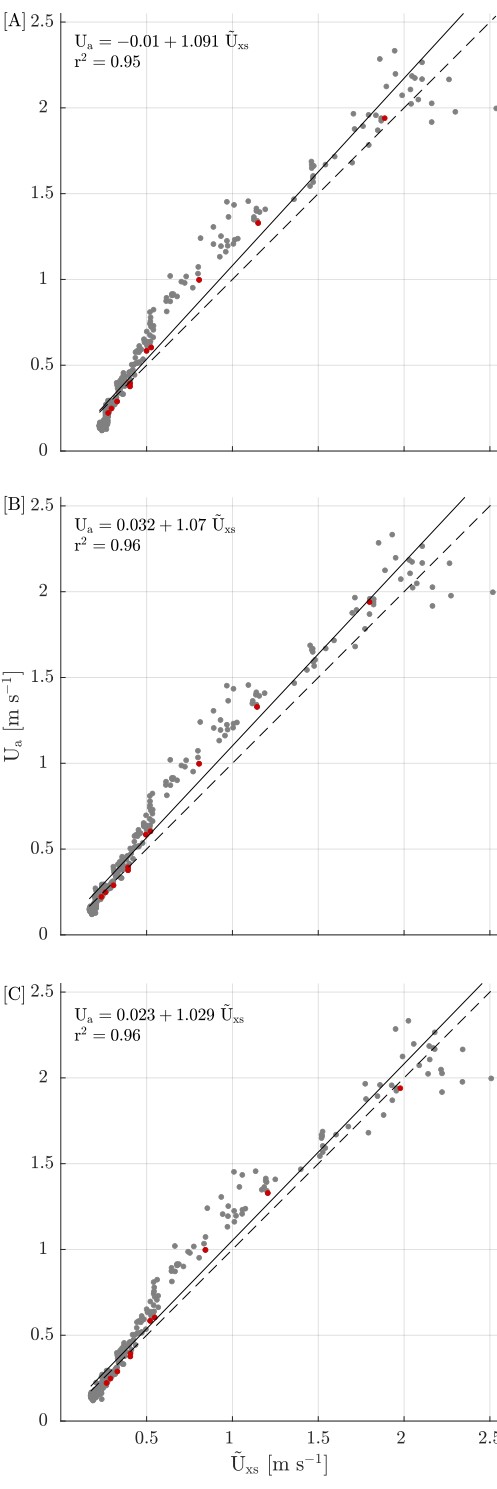

**Figure 4.** Relationship between the image-based 1-D flow velocity ($\tilde{U}_{xs}$) and 1-D flow velocity derived from Environment Agency flow gauging measurements ($U_a$). Interpolation between, and extrapolation beyond, observed cross-section velocities is achieved using [A] a quadratic function, [B] cubic function, and [C] constant Froude number assumption. Red dots are used to show the $\tilde{U}_a$ values that are plotted in Figure 5.

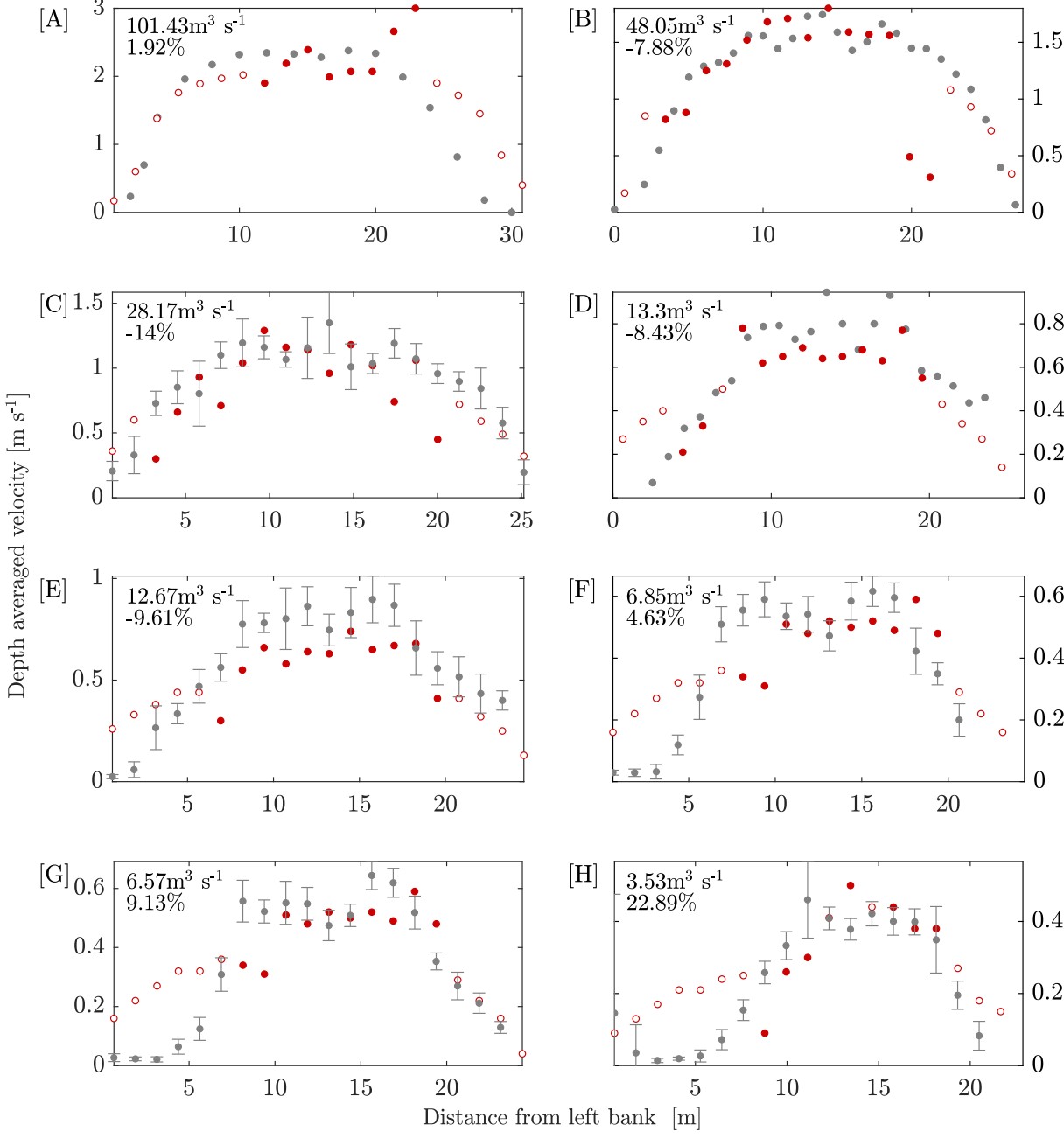

**Figure 5.** A selection of cross-section velocities illustrating the deviation between the Sontek aDcp velocity magnitudes (grey circles) and KLT-IV generated velocities for a range of flow conditions. Measured velocities are shown by the red filled circles, whereas estimates based on the constant Froude number assumption are shown by the red open circles. Error bars illustrate the standard deviation of stationary aDcp measurements. Flow discharge measurements for the aDcp transects are presented in each subplot along with the percent difference between discharge reported by aDcp and the reconstructed discharge using KLT-IV.





## 3.2 Application of an index-velocity

The hydrological conditions observed during the calibration period led to the retention of 214 flow gauging measurements

that were acquired at the same river stage ($\pm0.01$m) as the videos. This serves as a calibration dataset to enable the partial view of the camera to be accounted for. As a consequence of (i) the camera's field of view failing to capture the entire cross section, and (ii) surface velocities being reconstructed as opposed to the depth-averaged velocities, it was initially hypothesised that $\bar{U}_s \neq U_a$. To first explore the nature of this relationship, the deviation between $\bar{U}_s$ and $U_a$ is simulated. When only one $U_a$ is used to calibrate $\bar{U}_s$, the data indicates that $\bar{U}_s$ overestimates by 26%, however, there is a great deal of variability in

the outcome, with the IQR spanning 36% (Figure 6). As the number of $U_a$ used in the calibration increases, the variability is reduced, with the median percentage difference becoming stabilised when eight flow gauging measurements are used. In this scenario, the median output corresponds closely to that when 50 flow gauging measurements are used (27.4% vs 26.7%). This analysis indicates that $\bar{U}_s$ has a tendency to overestimate relative to $U_a$ and that the relationship between these two variables is influenced by the number of flow gauging measurements that are used to predict the relationship. However, in this instance it

would be possible to successfully quantify this relationship using as few as eight flow gauging measurements.

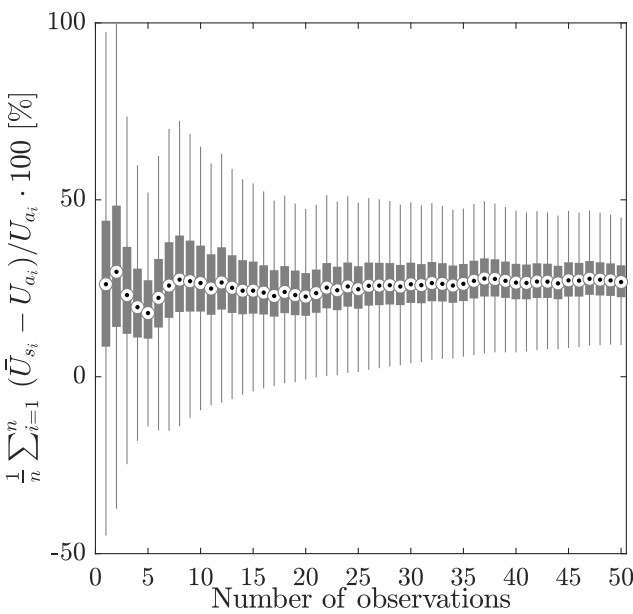

**Figure 6.** Results of monte-carlo simulations, where the number of paired selections of $U_a$ and $\bar{U}_s$ are varied to determine its influence on the calibration of the KLT-IV approach.

    This overestimation of $\bar{U}_s$ relative to $U_a$ is further highlighted when all 214 reference velocity measurements ($U_a$) are compared with the median of the distributed velocity measurements for the same river stage $\tilde{U}_s$. In this instance, a strong linear relationship ($r^2 = 0.96; p < 0.001$) can be observed (Figure 7a). However, $\tilde{U}_s$ overestimates by 16% on average. The calculation of the offset between $\tilde{U}_s$ and $U_a$ is subsequently applied to the $\tilde{U}_s$ measurements obtained during the validation





period (*n*=274), resulting in a much closer correspondence between the two variables ($r^2 = 0.98; p < 0.001$, Figure 7B). This finding indicates that the applied transformation developed during the calibration experiment holds true beyond that period with an acceptable level of uncertainty.

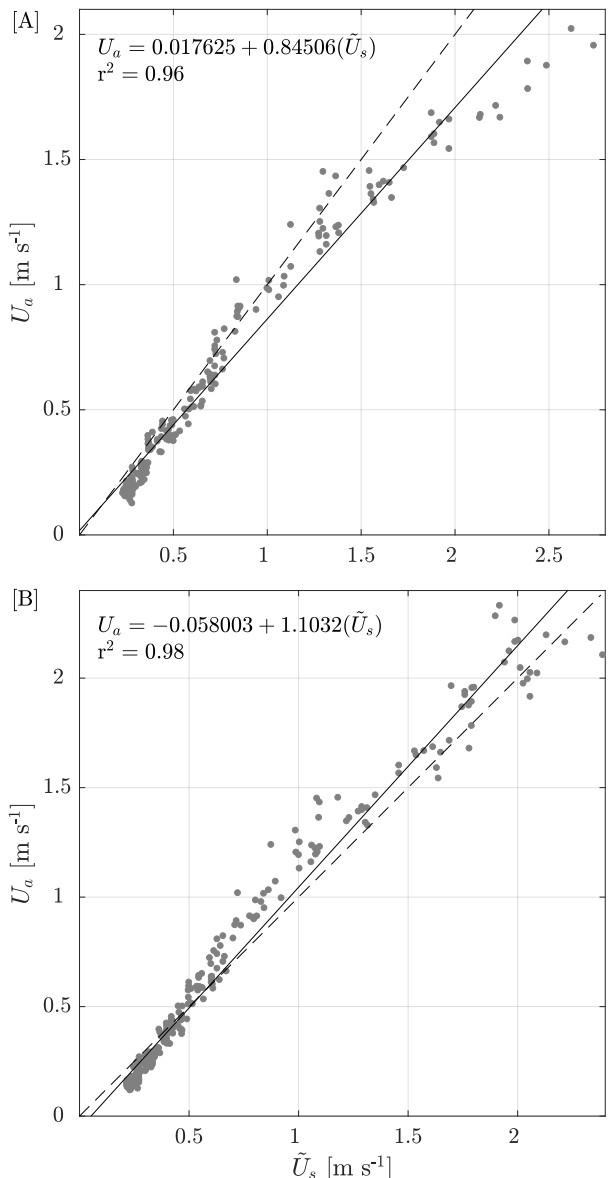

**Figure 7.** Bi-plot and linear fit between KLT-IV derived velocity measurements [$\tilde{U}_a$] and reference velocity measurements [$U_a$] during [A] the calibration period and [B] validation period. The solid line represents the linear fit between variables, with the 1:1 line also shown (dashed line).





## 4 Discussion

### 4.1 Application

Whilst this article has examined the inter-comparability of 1-D velocities obtained by image-based approaches, and reference measurements made via a variety of methods (e.g. electromagnetic current meter, aDcp, etc.), the utility of 1-D measurements obtained by image velocimetry techniques is likely to be in the development or refinement of stage-discharge rating curves. When we utilise the velocity data obtained by either the constant Froude number assumption or the distributed index velocity approach, we are able to generate discharge estimates that are broadly comparable with those generated by the standard Envi-
ronment Agency flow gauging approaches (Figure 8). When the relationship between image-based and reference gauging data is evaluated using a linear model with intercept of 0 and slope of 1, the coefficient of determination ($r^2$) values for the Froude number assumption and the distributed index velocity approach are 0.98 and 0.99 respectively, with Root Mean Squared Error (RMSE) values of 4.57 and 4.05 $m^3$ $s^{-1}$ respectively and a percent bias (PBIAS) of 5.5% and 3.4% respectively (Figure 8B). When it is considered that the reference data used represents significant efforts of hydrometry teams to make field measure-
ments in a range of challenging conditions between 1981-2019, and that the imagery used in this analysis was acquired for under one-year, and autonomously analysed in an unsupervised workflow, we can begin to identify the potential gains that wider employment of these techniques in appropriate environments may bring.

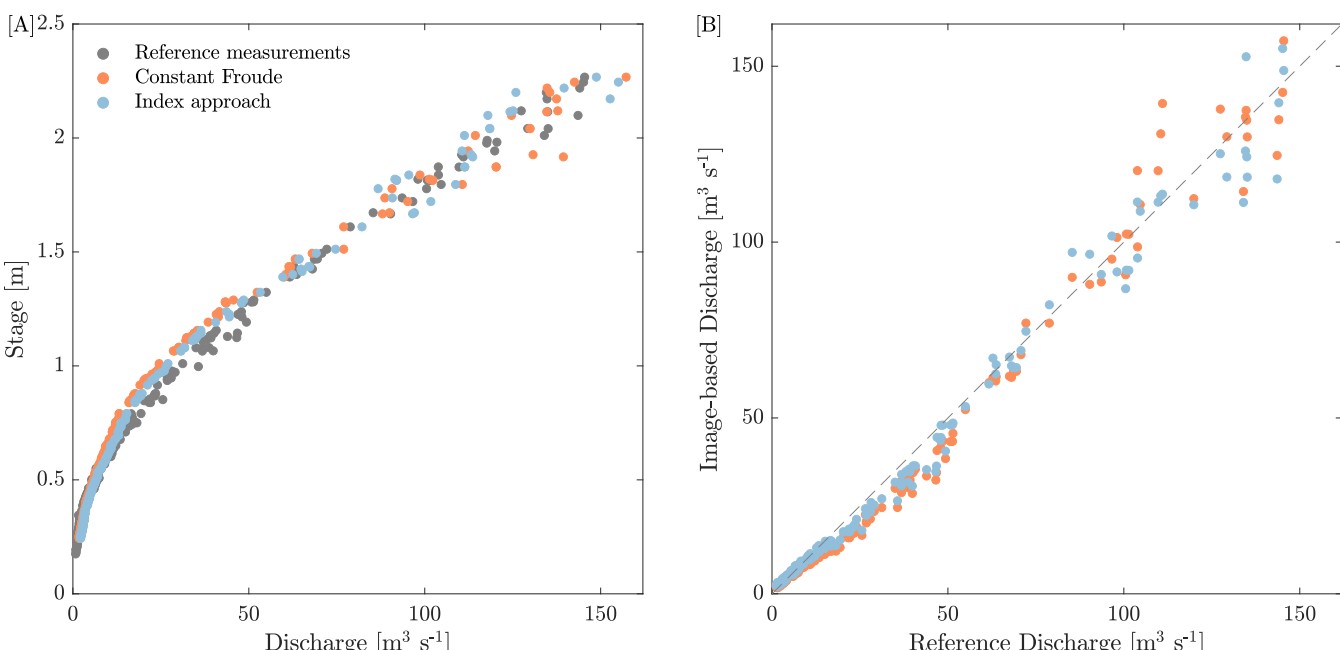

**Figure 8.** [A] Stage-discharge plots for the River Dart at Austins Bridge (UK) following analysis of imagery acquired over a one-year period (shown in the red and blue), and developed using conventional flow gauging techniques between 1981-2018 (grey). [B] Comparison between image-based discharge estimates and reference discharge estimates with 1:1 line shown.





## 4.2 Uncertainties in reference datasets

The comparisons conducted here have involved the computation of 1-D velocities using the observed $Q$ divided by the cross-
sectional area at the time of observation based on geodetic survey measurements. Our analysis indicates that over a nine year
period (2010–2019), the cross-section across the full range of flows varied by 5%. There is therefore a degree of uncertainty
concerning the stability of the cross-section across the time period for which reference gauging measurements were obtained
(1981-2018). Any variability will have a direct influence on the reference 1-D velocities calculated. In addition, the wetted
cross-section area calculated through the combination of geodetic survey and stage measurements has previously been found
to differ from aDcp derived cross-sectional areas, with cross-section average depths being underestimated by between 6-9.5%
(Kim et al., 2015). In the present study, the mean percentage error between these cross-section measurement methods is 7%
(12.5% for RDI StreamPro, 5% for Sontek RiverSurveyor M9) $\pm$ 10% (95% CI). This bias in aDcp cross-section measurements
produces uncertainty in the aDcp-derived flow measurements, given that the reported $Q$ is a product of the depth-cell size and
water velocity across the section (Mueller et al., 2009). In addition, given that the reference velocities used in this analysis
have been acquired from as early as 1981, there are inherent uncertainties in the acquisition methods adopted that cannot be
quantified.

## 4.3 Uncertainties in image analysis

Analysis of the aDcp transects enabled the computation of an $\alpha$ value relating surface velocity measurements to a depth-
averaged velocity. Analysis shows this value to be $0.87 \pm 0.07$ (95% CI), a value within the expected range for unmodified
channels consisting of a gravel bed (Turnipseed and Sauer, 2010). Given that this value is anticipated to vary as a function of
water depth or relative roughness it is of interest that no clear relationship between stage and $\alpha$ is observed. This complexity
exemplifies the importance of acknowledging the role of $\alpha$ in uncertainty assessments (Hauet et al., 2018).

    The videos analysed here are of 10-seconds duration, of which the first 3-seconds was discarded due to frame rate and
compression issues. This is a relatively short period of time for analysis to be undertaken with research illustrating the potential
benefits of analysing longer duration videos (Pumo et al., 2021), especially under poor seeding conditions (Dal Sasso et al.,
2018). Furthermore, if longer duration videos are available it may be possible to limit analysis to the image sequences with
optimal seeding characteristics, which can lead to potential gains in accuracy (Pizarro et al., 2020a, b).

    Given the requirement for this analysis to be unsupervised and automated, image enhancement was limited to application of
a high-pass filter only. Given the wide range of environmental and lighting conditions across the one-year monitoring period,
the visibility of the water surface and associated tracers will differ throughout. The choice of this procedure was to maximise
the visibility of potential tracers, however, this also comes at the risk of enhancing noise locally. In some instances, such as
when the river bed is visible under low flow conditions, additional image enhancement procedures would be beneficial e.g.
background subtraction. The choice of pre-processing procedures is dependent on the challenges that one is trying to resolve.
Therefore, implementation within an automated work-flow is non-trivial and the development of methods for generalisation is
worthy of further research.





In the generation of a camera model to be used for the generation of orthophotos which are subsequently analysed to determine feature displacement, an assumption is made that the water surface is planar. Under normal flow conditions this assumption is valid. However, under high flow conditions, surface waves of considerable height (» 10cm) develop resulting in variable water surface elevations throughout the domain. Given the perspective of the camera, these surface undulations may

result in biased flow velocity estimates due to part of the downstream component being registered as normal to the main flow line, or alternatively part of the normal component being registered as contributing to the downstream flow rate.

## 4.4 Outlook

Recent research has illustrated the precision of optical flow methods for reconstructing flow dynamics and highlighted their relative insensitivity to parameterisation (Pearce et al., 2020). This has naturally led to application of these techniques becoming

increasingly widespread for the purposes of obtaining dense surface flow fields, with their adoption into continuous monitoring workflows now becoming established (e.g. Hutley et al., 2023). However, a significant challenge in application of these methods within unsupervised workflows is the presence of environmental noise which leads to either a reduction in successfully tracked tracers or the presence of successfully tracked features that cannot readily be related to the depth-averaged flow. Generally this is presented as: i) noise that impacts the quality of the imagery and visibility of the water surface (e.g. in-homogeneous

lighting of the water surface, bright-spots on the camera lens, precipitation), ii) the water surface texture lacking sufficient detail to enable dense flow fields to be established; iii) environmental factors that influence the quality of the measurements (e.g. tracking of features affected by the presence of standing waves, or wind-induced effects). In the research presented here, we have not sought to address these issues directly. However, to improve robustness of the outputs, further analysis could be undertaken to identify and apply appropriate seeding density metrics to evaluate the quality of entire videos, or eliminate poorly

seeded image sequences from analysis, or focus analysis on specific cross-sections within the imagery to improve the quality of reconstructions. Alternatively, the influence of optical noise may be reduced through application of dynamic weighting across the image scene (e.g. Cao et al., 2022). Furthermore, utilisation of surrogate information (e.g. wind-speed and direction) may be used to identify time periods where wind-shear may have significant effects on the apparent surface velocities, allowing corrections to be established and applied where required. The potential for optical flow methods to be improved through

application of deep learning models is also significant. Ansari et al. (2023) provided evidence for a range of CNN optical flow models (collectively termed RivQNet) to improve flow reconstructions in challenging environmental conditions. Further refinement and training of these methods may offer significant performance benefits.

## 5 Conclusions

In this study, we investigate the potential for an open-source toolbox (KLT-IV) to reconstruct the surface flow field of the

River Dart (UK) for the purposes of estimating section averaged (1-D velocity) in an unsupervised and autonomous workflow. Given the partial view of the channel that is visible from the camera sensing system (73% of the channel width under normal flow conditions), application of appropriate data-fitting methods, or establishment of index-based approaches was required





to interpolate within and extrapolate beyond the field of view. Following image acquisition over a period of one-year, and following analysis of over 11,000 videos, we can draw the following conclusions:

1. Highly significant linear relationships ($r^2$ = 0.95-0.97) are established between reference 1-D velocities and those computed using KLT-IV in conjunction with data fitting techniques. The intercept of these models ranges from -0.01 to 0.032 and slopes range from 1.029 to 1.091 (Figure 4).

   2. The mean absolute percentage error in 1-D velocities (using the constant Froude assumption) relative to those produced using a SonTek River Surveyor M9 aDcp is 9.8% (Figure 5).

3. An index-velocity approach is developed which relates the mean of the observed flow field to the reference 1-D velocity. The form of this relationship was established during a calibration period spanning March to June 2018, and this was subsequently applied and tested for a validation period (July 2018 to March 2019). In the validation period, a highly significant linear relationship ($r^2$ = 0.98) was obtained between the mean values of the flow field and the reference 1-D velocities (Figure 7).

4. We use the best performing data-fitting approach and index-based approach to estimate flow discharge at the monitoring site. When these are compared with the reference data obtained by the Environment Agency, $r^2$ values of 0.98 to 0.99 are obtained (for a linear model), with a percentage bias of between 3.4% and 5.5%, respectively (Figure 8).

   5. We identify uncertainties in both the reference datasets and image-based analysis that may be of significance. For example, in the case of the reference data we identify a bias in aDcp-based cross-section measurements relative to those made
using geodetic surveys of 7%; and in the case of the image-based analysis, we identify an $\alpha$ of $0.87 \pm 0.07$ (95% CI) following analysis of aDcp profiles. However this does not vary in a systematic way, which may influence the resulting conversions from surface velocity to a depth-averaged velocity.

   6. This approach is well suited to being used in operational, and real-time settings. This is due to the relatively few parameters that must be defined following initial set-up. In the present analysis all parameters are kept constant and not varied
as a function of river stage. However, a thorough assessment of the dependency of flow field reconstructs with varying environmental and hydro-geomorphic conditions remains a research gap.

*Code and data availability.* The version of KLT-IV used for this analysis can be found at https://github.com/CatchmentSci/KLT-IV. Data used in the production of this article can be accessed at Perks (2024a). Scripts used to generate the Figures presented in this article can be accessed at: https://github.com/CatchmentSci/automated-computation-of-river-flow-velocities.





*Author contributions.*  MTP led the investigation including conceptualisation, formal analysis and writing; BH contributed to methodology, and original draft preparation; JR provided resources, specifically access to the UK Environment Agency datasets, all authors contributed to project conceptualization, review and editing.

*Competing interests.*  The authors declare that they have no conflict of interest.

*Acknowledgements.*  This work was funded by NERC grant NE/K008781/1 "Susceptiblity of catchments to INTense RAinfall and Flooding
(SINATRA)", and in collaboration with the Environment Agency. The authors thank the Environment Agency for providing the flow time-series, gauging records, and cross-section information for the River Dart at Austins Bridge (Station number: 46003).





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
