# Peer review of "Unsupervised image velocimetry for automated computation of river flow velocities"

_Hydrology and Earth System Sciences, 2024_

## Author Response (AR1)

**Response to Reviewer 1**

**General Comments**

**Reviewer Point P 1.1** — The manuscript deals with image processing aimed at obtaining reliable flow discharge estimates under continuous unsupervised operation. The utility of this approach is acknowledged by the WMO, recently issuing an update to WMO-No. 168, section 5.3.7.3 – Image-based methods for discharge measurements. The authors add to the general discussion on the applicability, accuracy and associated uncertainties by presenting the results from nearly one-year continuous observations on the River Dart, UK. The data were collected at Austins Bridge, where available hydrometric records span back to 1958, and treated in compliance with the WMO-No. 168 requirements.

The River Dart at Austins Bridge yields the longest hydrometric record on the River Dart, but its hydrologic presentation in subsection 2.1 is apparently scarce. It seems however important to put the experimentation in the proper context, in terms of both average and extreme flows. The notion on 'reference observations' appears as early as in Subsection 2.5.1, where they are briefly mentioned in L172 as a source of the conversion factor, but the details are lacking. What is their distribution across the velocity/discharge range? What are the standard data collection protocols? Even the number of observations is uncertain. The total number of reference observations (303) is presented in the Abstract, but never discussed elsewhere. Were some of the, or all aDcp observations (at least some were, as stated in L182, but clearly not all, since the record starts is 1981)? Subsection 2.5.1 is vague on the subject, L177 has no notions on the Ua protocol, and in the rest of the subsection only aDcp data are discussed.

The manuscript is generally well-written, and most issues that can be reported are related to an overall quality of presentation and text flow. While the manuscript grosso modo has most of the data needed, some of these data are scattered across the manuscript making it hard to follow.

**Reply**: Thankyou for the constructive comments. We implement each of these suggestions in the revised manuscript, and below outline how this is achieved.

With reference to the point concerning the lack of context, information about the location where the experiment takes place, and information about the reference measurements used, we have sought to address these concerns. Specifically, in the revised document, *Section 2.1 Experimental Site* deals with a more complete description of the geographical and hydrological background of the site. Furthermore, an additional section is provided, *Section 2.2 Reference Data*, which provides detailed summaries of the reference gauging data and information on how this is used to establish Ua. We also provide reference to the flow measurement data acquired by the Environment Agency, which we use in our analysis. This consists of summary information documenting the instruments and methods used to acquire the reference gauging data.

**Specific Comments**

**Reviewer Point P 1.2** — L31: are these mentioned below algorithms, in the sense of Kanade-Lukas algorithm, or rather methods and approaches, as per WMO-No. 168?

**Reply**: These are perhaps better described as classifications of methods (per WMO-No. 168). This has therefore been revised to read: *'Measurements of surface velocity can be achieved through the application of methods that may be broadly categorised as: particle tracking velocimetry (PTV) (Brevis et al., 2011; Tauro et al., 2017); large scale particle image velocimetry (LSPIV) (Fujita et al., 1998; Muste et al., 2008); or space-time image velocimetry (STIV) (Fujita et al., 2007)'.*

**Reviewer Point P 1.3** — L41: reference the Lukas-Tomasi algorithm, and probably mention Kanade here also, since KLT in KLT-IV might presumably stand for 'Kanade-Lukas-Tomasi' (indeed, references from L132 can be moved up here).

**Reply**: This suggestions, in addition to Reviewer 2's comments on this sentence, have been implemented in the revised document to read: *'These computer vision algorithms e.g. Kande-Lucas-Tomasi algorithm (Lucas and Kanade, 1981; Tomasi and Kanade, 1991; Shi and Tomasi, 1994), automatically identify pixels that are distinct from their neighbours, and these distinct features can be iteratively tracked through a sequence of images'.*

**Reviewer Point P 1.4** — L43: PIV is not presented above, but is apparently the same as "PIV" in LSPIV, for the latter is the same PIV in the narrower context (WMO-No.168).

**Reply**: in the revised version of the manuscript we replace mention of PIV with the more specific LSPIV term. The revised sentence reads: *'These approaches are computationally very efficient and capable of performing analysis up to two orders of magnitude faster than traditional LSPIV and PTV approaches (Tauro et al.,2018b)'.*

**Reviewer Point P 1.5** — L43-44: in referencing (Tauro et al., 2018b) it can be noted that at least some of the results of that paper were obtained using "woodchips . . . continuously deployed" from the upstream so not fully in the context of a "continuous, automated, and unsupervised workflow".

**Reply**: This is correct. We were using reference to this particular paper to illustrate that the results generated by optical flow methods are comparable to standard technologies, rather than suggesting that Tauro et al (2018) tested the algorithms in a continuous and automated workflow.

**Reviewer Point P 1.6** — L94-95: this phrase evokes the question on what were the hydrological conditions during the experiment, of which there is no notion in Section 2.1. How they are compared to long-term averages and extremes? It is only later in the manuscript (L178) that we will know that the reference flow measurements range from 1.7 to 145 m3s-1; no info is provided on the flow discharge ranges during the experiment.

**Reply**: We have addressed this comment through the addition of new material provided in Sections *2.1: Experimental Site* and *2.2: Reference Data.*

**Reviewer Point P 1.7** — L170: '20 segments of equal size'; from Figure 2, these are segments of equal width, not equal area, though for polygons the latter is most commonly assumed. This needs to be stated explicitly. Am I right to understand this so that while the flow width increases with the increasing discharge, the width of each segment increases accordingly?

**Reply**: Very true. These segments are of equal width, not equal size and this has been modified in the revised document. Yes, as the flow width increases, the width of each cell will increase to maintain the condition of 20 segments of equal width.

**Reviewer Point P 1.8** — Also, this first paragraph of the Subsection 2.5.1 is somewhat confusing. First, while the basic references are provided, I would be interested in having more details on how exactly the observed data were extrapolated over the regions out of the camera sight using these three techniques; why there are 20 segments, and not more/less – how is this justified? In L170, was it so that "average velocities for 20 segments of equal sizes are established" stands for the water surface velocities averaged across the segment width? What is the number of pixels with data in each segment surface? How exactly was alpha derived? The (Perks, 2024b) reference is misguiding in that might presumably lead to the supportive information while it simply references a figure derived from unexposed data. Were the alpha values derived segment-wise for all 60 aDcp measurements? Were the segments similar or close in size (width?) in aDcp and extrapolation exercises? Further on, is the "cell" in L173 equivalent to "segment" in the lines above? If not, the difference must be explained otherwise the text is ambiguous.

**Reply**: To apply the constant Froude number approach, linear regression between cell depth and cell average velocity was performed with the model intercept constrained to zero. For cells with missing data, velocities were estimated by multiplying the cell depth with the slope of the linear function. Where quadratic or cubic polynomials were used to estimate velocities in cells with missing data, data fitting was performed using the paired measurements of distance along the section and cell averaged velocity, with the addition of velocity values of zero at the channel boundaries. Cells in the section with missing values were subsequently estimated using the obtained quadratic or cubic polynomial function. This information has been added to Section 2.6.1.

In terms of the selection of 20 cells, we select this value so that our discharge calculation may meet the guidelines set out by ISO 748 which state that the discharge of any verticals should not be more than 10 per cent of the total discharge and ideally no more than 5 percent of total expected discharge, and that in fixing verticals, equal spacing should be preferred wherever possible.

The number of velocity measurements present within each cell may be as low as zero (requiring interpolation/extrapolation), single digit numbers, up to several hundred. With an increasing number of successfully tracked tracers within a particular cell, the greater the level of confidence in the reported average values. However, a cell with a single velocity reconstruction may also be valid.

Alpha is derived by producing a power-law fit between the normalised river velocity and normalised distance from the river bed using all aDcp pings for an individual transect. Using a unique function developed for each transect, we calculate the ratio between the (extrapolated) velocity at the water surface with the predicted velocity at 0.6D (which corresponds to the theoretical depth-averaged velocity). This ratio provides us with the alpha value for each individual transect. Finally, we take the median of all these alpha values and employ this in the scaling of surface velocities to depth-averaged velocities. In addition to the summary plot that we initially provided, we have also added plots showing the alpha results for each individual transect and provide reference to this data in the main text (Section 2.6.1).

In the original submission we used the terms 'segment' and 'cell' interchangeably and acknowledge this as an error. In the revised version of the manuscript we consistently use the term 'cell' to describe the 20 individual units that make up the cross-section area.

**Reviewer Point P 1.9** — L180: In "calculate the (1-D) velocities derived from reference observations", can these be replaced simply by Ua, as given in L177? Similar to this, can "1-D" be omitted from most instances where its presence is misleading or redundant, e.g., L184, L187, L191-192, L198 and almost each and every instance below. Once the definition is given, there is no need to reiterate, and it makes sense to substitute. As in L180, same in L196-197, same in L198-199, and so on.

**Reply**: Thank-you for this comment. In our initial submission we intentionally reiterate the definition throughout based on feedback on an earlier draft. Previous feedback suggested that use of Ua and Us may make the text more difficult to follow. However, given your feedback we will remove the redundant messaging as outlined in your comment.

**Reviewer Point P 1.10** — In this Subsection 2.5.2, again, the statistics for reference stage observation can be highly utile. Were these reference observations mentioned in L196 the same as presented elsewhere including the Abstract? Can the authors consider providing the details either here or in the Section 3?

**Reply**: In Sections 2.6.1 and 2.6.2 we now add additional information about the flow range of the reference measurements that are used in the calibration and validation phases of the research.

**Reviewer Point P 1.11** — L210: It is probably more correct to not start the "Velocity reconstruction" section with the overview of the EA data. Also, in Figure 3, velocity reconstruction is applicable to both procedures employed, see Subsections 2.5.1 and 2.5.2 accordingly. Thus Section 3.1 must be renamed accordingly.

**Reply**: Thank-you for this suggestion. This information has now been placed in the newly produced Section 2.2 (Reference Data).

**Reviewer Point P 1.12** — L220: more justification is needed here or above on claiming the constant Froude as a best-performing model, since the Figure 4 is indecisive on this matter.

**Reply**: Additional detail is provided in this section to illustrate that at flow velocities in excess of 0.59 m s$^{-1}$ the Froude approach produces velocity estimates in closest agreement with the reference velocity.

**Reviewer Point P 1.13** — Figure 5: is it correct that the distances are given from the left bank while the camera is on the right bank?

**Reply**: Yes this is labelled correctly.

**Reviewer Point P 1.14** — L244-245: how exactly the conclusion that "as few as eight flow gauging measurements" suffice to successfully quantify the cross-section averaged velocity was reached? Does it imply that the uncertainty (variability) level is acceptable at this n.

**Reply**: In the modified version of the manuscript, this is rephrased to highlight that the levels of uncertainty do not appreciably change when the number of measurements used in the calibration is increased beyond eight.

**Reviewer Point P 1.15** — L266: "autonomously analysed in an unsupervised workflow". This said, it must be noted however that while the imagery was indeed collected in that way, its further treatment and the final output weren't possible without the prior work on-site, including aDcp surveys under challenging conditions of the below 1% flood. Without this, based solely on the a priori knowledge of the site, what the potential errors of the standalone application could have been?

**Reply**: This is a fair point. In order to adopt one of the theoretical flow distribution approaches in the autonomous workflow we needed to have cross-section information and an estimate of the $\alpha$ value to convert the surface velocities to a depth-averaged. To adopt the index-velocity approach in the autonomous workflow we would need to have a number reference flow gauging's to calibrate the surface velocities to. We have added the following text to Section 4.1 to clarify the requirements of the unsupervised workflow: *'The particular utility of this approach is that image velocimetry analysis can be conducted in an autonomous environment following camera calibration, with inputs of a water level timeseries that correspond to the time of video acquisitions. Additional information such as the cross-section geometry and an estimate of $\alpha$ to convert water surface to depth-averaged flow velocities, or information relating the surface velocity from across the domain with a section-averaged velocity is also required'.*
  Whilst analysis of aDcp transects allowed calculation of the site-specific $\alpha$ of 0.87, experience shows that this value tends to approximate 0.85 in similar fluvial settings. Therefore in this instance an estimate of $\alpha$ would yield quite comparable results. However, for the application of the index-velocity approach in this particular study, the calibration procedure is required to obtain accurate estimates (Figure 7).

**Reviewer Point P 1.16** — L278-279: Top and bottom blind zones are present in aDcp measurements, along with right and left margins, so the reported Q is corrected for these in RiverSurveyor software. The wetted cross-section area has some effects on the aDcp-derived flows, but it is not to be overestimated.

**Reply**: The statements that this comment refers to are qualified in the revised document, with removal of the text: *'This bias in aDcp cross-section measurements produces uncertainty in the aDcp-derived flow measurements, given that the reported Q is a product of the depth-cell size and water velocity across the section'.*

**Reviewer Point P 1.17** — Overall, I concur with the authors in that the major utility of the described approach is in the "development or refinement of stage-discharge rating curves", where the OTV and OTV-like approaches can be highly utile in increasing the accuracy of high flow estimations. For the River Dart at Austins Bridge, the highest observed peak flow is 550 m3s-1 (1979-1980), almost 400 m3s-1 over the observed highest peak flow. At this part of the flow duration curve, as far as I understand, the OTV can substantially improve the flow estimates in the straight single-thread non-deformable channel.

**Reply**: Thank-you for the constructive comments made within this review. We believe that the manuscript will be significantly improved by addressing these as outlined in the responses above.

**Response to Reviewer 2**

**General Comments**

**Reviewer Point P 2.1** — This paper presents an interesting study on the use of a camera gauge for long-term measurement of flow velocity and discharge. Its structured and clear approach, coupled with the evaluation of an automatic system over a period of approximately one year, is a noteworthy contribution. Also, the discussion of interpolating and extrapolating surface velocities demonstrates the potential of camera gauges as a viable supplement or maybe even alternative for continuous monitoring in hydrology. However, I think, there are several areas where further elaboration or clarification is needed. The paper focuses primarily on the methodological aspects, and therefore, I think it would be more suitable to be considered as a technical note rather than a research paper. This reclassification would better reflect its contribution.

**Reply**: Thankyou for the constructive comments. We implement each of these suggestions in the revised manuscript, and below outline how this has been achieved. With regards to the placement of the article as 'Research article', or 'Technical Note' I will take further advisement on this decision from the Editor. However, from a personal perspective, although the article is indeed technical in nature, we hope that the contribution goes beyond what would be expected of a 'Note' and that this could be considered as a 'Research article' contribution.

**Reviewer Point P 2.2** — The authors assume stable intrinsic and extrinsic parameters throughout the year-long observation period. This assumption warrants deeper discussion, particularly in light of potential influences like temperature changes (e.g. as shown by Elias et al., 2020) or environmental factors like wind causing camera movement. Would a frequent update of the position/orientation not be needed? What are the expected error ranges if no update occurs? Addressing these issues in more detail is necessary to increase the robustness of the conclusions.

**Reply**: This point about our assumption of stable intrinsic and extrinsic parameters is an interesting one. Firstly I would like to address the extrinsic parameters (related to the stability of camera orientation):

We agree that a stable frame of reference is critical when deploying camera systems in an automated workflow across prolonged deployment periods. Without the continuous presence of GCPs, we rely on calibrating the extrinsic camera parameters at a single point in time (at the start of the monitoring period). We have performed some analysis on the intra-image stability of the camera orientation between the start (March 2018) and the end (January 2019) of the monitoring period (Figure 1). When we compare clearly visible points (corners of the stage boards located on the far bank), we can identify that the pixel coordinates of the edges are comparable between images (within 1-2px). Using the point cloud survey obtained on-site at the time of camera setup we can estimate that the width of the gauge board is 22cm. Therefore imagery offsets at this distance are likely to be in the order of 1–3cm, which is within the general uncertainty of the registration process. We incorporate these findings in the revised manuscript (Section 4.3), and advocate for assessments such as this as standard practice. Furthermore, we highlight the benefits of permanent GCP networks in deployments that permit which would enable a time-series of extrinsic

parameters to be established (Section 4.3). In terms of the stability of image sequences within a single video, the impacts of small frame-to-frame movements may have an impact on the quality of velocimetry reconstructions and we do not seek to quantify this in this article. The role of external environmental conditions (wind, rain) on reconstruction quality is certainly a valid and open question in the field, but this is beyond the scope of the current investigation. Our method of analysing multiple videos for a given flow stage and adopting the median of the 1D velocity estimates for comparison purposes seeks to minimise the effects of external environmental factors which may affect the quality of the velocity field reconstruction for individual videos.

[Figure]

[Figure]

Figure 1: Testing the consistency between images acquired on the first and last day of analysis. Note: the [X,Y] labels within the imagery describe the pixel locations of the stage board edges.

With regards to the stability of intrinsic parameters, this is not something that we have considered in our analysis and as far as I am aware this has not been assessed with reference to the application of image velocimetry. Prior to deployment, the camera calibration coefficients were established in a lab setting using a checkerboard, but we have not assessed the influence of external environmental characteristics on this. This is an interesting point and we thank-you for bringing this to our attention. We have provided some discussion around these elements in *Section 4.3: Uncertainties in image analysis* of the revised manuscript.

**Reviewer Point P 2.3** — The assumption of stable flow conditions during the calibration period also needs further exploration. I think, it would be beneficial to discuss how this impacts measurement accuracy, particularly over extended periods with potential varying riverbed or cross-sectional conditions. The authors could include further metrics such as standard deviations and deviations from the reference measurements temporally resolved during the observation period to provide more insight of such influences.

**Reply**: Significant changes to the riverbed and cross-section morphology could indeed have impacts on the Q (and resultant 1-D velocity) estimates that we produce and the potential for this source of

error is highlighted further in *Section 4.3: Uncertainties in image analysis* of the revised manuscript. In this particular case, whilst there is evidence for significant geomorphic change at this site (in 1979), one of the reasons that this location was selected due to its stable rating curve over the intervening years. Indirectly, the presence of a stable rating curve would provide evidence for the cross-section also being stable. Furthermore, as reported in the research article, our analysis of repeat surveys between 2010 and 2018 indicate that the cross-section has changed by no-more than 5% across the full range of flows experienced. We have since discovered an additional geodetic cross-section survey conducted in October 2020, which further illustrates the stability of the channel cross-section at this location (Figure 2). The differences in profile elevations between 5-10m on the x-axis is a consequence of a new cableway stanchion being built (Figure 3). This was erected after the field experiment was completed. All of the data available to us indicates that morphological change at the location of the field experiment is negligible and therefore any potential variability in measurement accuracy over time is likely generated by other sources. However, we do acknowledge that cross-sectional variability may be a significant factor for some locations.

[Figure]

Figure 2: Cross-section survey data at the experimental site acquired in 2018 and 2020.

[Figure]

Figure 3: Image showing the presence of a new cableway stanchion installed in February 2020 (after the end of the field experiment documented here).

**Reviewer Point P 2.4** — This study highlights the great potential of camera gauges for long-term, nearly continuous monitoring of hydrometric parameters. However, further discussion of the methodological assumptions and calibration limitations would greatly enhance its applicability.

**Reply**: Thank-you. We hope that our additional assessment of the stability of extrinsic camera parameters, and time-series analysis helps to allay any concerns about the apparent methodological assumptions that we make in the research article.
* * *
**Further smaller comments**

**Reviewer Point P 2.5** — L38-41: The description of the Lucas-Tomasi method should clarify that it is not a stand-alone approach to Particle Tracking Velocimetry (PTV) or Particle Image Velocimetry (PIV). Rather, it is another matching algorithm besides, e.g., NCC or FFT based approaches – however with the advantage of being a least square approach allowing for particle rotation and deformation during tracking, that complements particle detection methods or grid-based strategies.

**Reply**: Following the comments of Reviewer 1 and Reviewer 2 in relation to this sentence, we have altered this text so that the focus is on optical flow methods generally rather than to a particular algorithm. The text has been revised to read: *These computer vision algorithms e.g. Kande-Lucas-Tomasi algorithm (Lucas and Kanade, 1981; Tomasi and Kanade, 1991; Shi and Tomasi, 1994), automatically identify pixels that are distinct from their neighbours, and these distinct features can be iteratively tracked through a sequence of images.* More specific information regarding the detection and tracking method employed within the current study can be found in *Section 2.5: Image Processing.*

**Reviewer Point P 2.6** — L104: Clarify what is meant by a 'distorted camera model'. Do the authors mean a camera model that describes the intrinsic camera parameters including distortion parameters?

**Reply**: Yes, by distorted camera model we mean a camera model that describes the intrinsic camera parameters including distortion parameters. In *Section 2.4: Image Calibration* of the revised document we clarify this by removing mention of the 'distorted camera model' and instead state that: *'a site-specific camera model is developed to mathematically describe intrinsic parameters (e.g. focal length, lens distortion), and external parameters (e.g. location and orientation)'.*

**Reviewer Point P 2.7** — L107: The role of location and orientation (iii) in the workflow needs clearer distinction. Are these parameters derived from ground control points (GCPs) via spatial resection? This should be elaborated for better understanding. Is the location/orientation not the consequence from using the GCPs measured in the image and in the object space and then estimating the location/orientation using, e.g., spatial resection?

**Reply**: Within the KLT-IV software the user may choose to manually fix the camera's position and/or orientation, in which case these parameters will not be allowed to vary in the optimisation process. However, under normal use (and in this particular case), the user would simply provide an

estimate of the camera location and orientation. These free parameters would then be optimised with the use of GCPs as you allude to in your comment.

We have sought to clarify this in the revised document by stating that: *'Within KLT-IV the user must provide: (i) surveyed location of ground control points (GCPs); (ii) initial estimates of the camera location [x, y, z] and orientation; and (iii) the known height of the water surface being sensed. The user may also provide the intrinsic parameters of the camera if known.'* In terms of the specific approach used in the optimisation of the camera model, later on in this section we write that: *'The camera location [x, y, z] and view direction [yaw, pitch, and roll] was initially estimated using the point-cloud survey. These characteristics were then defined as free parameters and optimized to minimize the square projection error of the GCPs using a modified Levenberg–Marquardt algorithm'*.

**Reviewer Point P 2.8** — L120: Confirm whether the projection error refers to deviations in object space or image space, i.e., do you indeed mean projection error or rather reprojection error?

**Reply**: Thank-you for this comment, the method seeks to optimize the camera by projecting the GCP world coordinates to image coordinates and comparing the fit to the GCPs in image space [px]. My understanding is that this would be termed projection error.

**Reviewer Point P 2.9** — L163: There is already a study showing that one surface velocity from image velocimetry is enough to get discharge with entropy approach (Bahmanpouri et al., 2022)

**Reply**: Thank-you for brining this to our attention, we include reference to this work in the revised manuscript.

**Reviewer Point P 2.10** — L228-232: What is the reason for the deviation?

**Reply**: Greater error estimates in the near field of the imagery under the highest flow conditions are likely to be a consequence of the water surface deviating from our planar assumption. Due to camera orientation, water surface undulations will have a greater impact on the velocity reconstructions in the near-field than in the far-field of the imagery. In the revised version of the manuscript we make reference to *Section 4.3*, where we discuss possible uncertainties in the image analysis process.

**Reviewer Point P 2.11** — L245: The distribution of the eight reference measurements across different water levels should be clarified. It is critical to evaluate whether these measurements adequately represent the range of conditions encountered at the specific study site. Might it be more important to consider range of water level at which reference available and how well this represents most gauge situations?

**Reply**: The statement made in Line 245 or the original manuscript has been removed in the revised version. We also provide further clarification on the sampling protocol in Section 2.6.2, where we have added: *'Values of the samples selected for each step of the calibration (n = 1–50) are conditioned by the frequency distribution of the gauging data, with the likelihood of sampling a particular value being dependent on its frequency in the population. Inevitably, few flow gauging measurements are obtained at the highest flow magnitudes, with the median of the gauged discharge values being 6 $m^3 s^{-1}$. Therefore, the calibration sample protocol reflects the distribution of the actual flow gauging record'*.

**Reviewer Point P 2.12** — L304: The authors can estimate the error influence theoretically by varying the water level and flow velocities and see how strong changes of estimated velocities are.

**Reply**: Absolutely, and this is an active area of research for us. We are currently working to see how altering representation of the water surface may be used to refine velocity reconstructions when the assumption of a planar surface is violated. This analysis is currently under way and will be published as a separate research article.

**Reviewer Point P 2.13** — References – Note, please do not consider the references as a suggestion to be implemented in the manuscript rather than a proposal to more literature to underline my comments made: Elias, M., et al. (2020): Assessing the influence of temperature changes at the geometric stability of low-cost chip cameras. Sensors, (https://www.mdpi.com/1424-8220/20/3/643) Bahmanpouri, F., et al. (2022): Estimating the Average River Cross-Section Velocity by Observing Only One Surface Velocity Value and Calibrating the Entropic Parameter. Water Resources Research, (https://doi.org/10.1029/2021WR031821)

**Reply**: Thank-you for providing these references.